# Extracellular spreading of Wingless is required for *Drosophila* oogenesis

**Xiaoxi Wang**<sup>☯</sup>, **Kimberly S. LaFever**[ID]<sup>☯</sup>, **Indrayani Waghmare**[ID]*, **Andrea Page-McCaw**[ID]*

Department of Cell and Developmental Biology, Program in Developmental Biology, Vanderbilt-Ingram Cancer Center, Vanderbilt University School of Medicine, Nashville, Tennessee, United States of America

☯ These authors contributed equally to this work.

* indrayani.waghmare@vanderbilt.edu (IW); andrea.page-mccaw@vanderbilt.edu (AP-M)

**Data Availability Statement:** All relevant data are within the manuscript and its Supporting Information files.

**Funding:** This work was supported by R01 GM117899 to A.P.M. from the National Institute of

## Abstract

Recent studies have investigated whether the Wnt family of extracellular ligands can signal at long range, spreading from their source and acting as morphogens, or whether they signal only in a juxtacrine manner to neighboring cells. The original evidence for long-range Wnt signaling arose from studies of Wg, a *Drosophila* Wnt protein, which patterns the wing disc over several cell diameters from a central source of Wg ligand. However, the requirement of long-range Wg for patterning was called into question when it was reported that replacing the secreted protein Wg with a membrane-tethered version, NRT-Wg, results in flies with normally patterned wings. We and others previously reported that Wg spreads in the ovary about 50 μm or 5 cell diameters, from the cap cells to the follicle stem cells (FSCs) and that Wg stimulates FSC proliferation. We used the *NRT-wg* flies to analyze the consequence of tethering Wg to the cap cells. *NRT-wg* homozygous flies are sickly, but we found that hemizygous *NRT-wg/null* flies, carrying only one copy of tethered Wingless, were significantly healthier. Despite their overall improved health, these hemizygous flies displayed dramatic reductions in fertility and in FSC proliferation. Further, FSC proliferation was nearly undetectable when the *wg* locus was converted to *NRT-wg* only in adults, and the resulting germarium phenotype was consistent with a previously reported *wg* loss-of-function phenotype. We conclude that Wg protein spreads from its source cells in the germarium to promote FSC proliferation.

## Author summary

Wingless (Wg)/Wnt proteins act as important signals between cells in many contexts. For decades, studies in the *Drosophila* wing established that Wg signals to distant cells, implying that Wg spreads extracellularly. However, studies in other tissues and organisms have found Wnt ligands signal in a juxtacrine manner, to neighboring cells. Recently the importance of Wg spreading was re-evaluated in the fly wing, spurred by the finding that membrane-tethered Wg, unable to spread from its source cell, can substitute for Wg. These findings fueled a search for other tissues where Wg extracellular spreading is required. The nature of Wg signaling in *Drosophila* oogenesis has been unclear. In the germarium a visible gradient of Wg spans ~50 μm, reaching from its source to follicle stem

General Medicine, https://www.nigms.nih.gov. I.W.
was funded by 5T32CA119925-12 from the
National Cancer Institute, https://www.cancer.gov.
The funders had no role in study design, data
collection and analysis, decision to publish, or
preparation of the manuscript.

cells, but it has been argued that Wg signals from neighboring cells to the stem cells. In
this study, we tested the role of Wg spreading by analyzing oogenesis in the tethered-Wg
flies. Two copies of tethered Wg cause a non-specific Wg toxicity; however, even when the
dose tethered Wg is reduced, ovaries have negligible follicle stem cell proliferation and
produce few eggs. Thus, extracellular Wg spreading is essential for follicle stem cell prolif-
eration and oogenesis.

## Introduction

Wnt signaling is an important and conserved mechanism of cellular communication that con-
trols proliferation and differentiation in many cell types and organisms [1]. The first Wnt
ligand identified in any animal was encoded by the *wingless (wg)* gene in *Drosophila melanoga-
ster* [2], and subsequent studies of the Wg protein have provided important paradigms for its
function in all animals. Beginning over 25 years ago, accumulated studies have established that
Wg meets the criteria for a morphogen: it spreads in a gradient from its source exposing cells
to different concentrations of the protein, and it triggers different transcriptional responses at
different concentrations [3–9]. This morphogen model is based largely on studies in the wing
disc, an immature larval structure (anlage) that gives rise to the adult wing. More recently,
however, the morphogen function of Wg was significantly challenged by the finding that flies
are viable and patterned normally when the gene encoding the secreted Wg ligand was
replaced with a membrane-tethered version of Wg, NRT-Wg, capable of signaling but not
spreading from its source [10]. Although these flies were reported to be less healthy than their
control siblings, their wings were patterned normally, indicating that Wg extracellular spread-
ing is not important for wing pattern. This highly influential study was interpreted to mean
that the observed spreading of Wg protein is not important for its signaling, as Wg signaling
occurs only in a juxtacrine manner between adjacent cells rather than spreading in a diffusive
manner from source cells to target cells [11].

Several groups have re-examined the question of extracellular spreading of Wg using the
*NRT-wg* allele. Two studies have found that Wg protein signals at a distance–requiring extra-
cellular spreading—from gut tissues. In embryos, NRT-Wg cannot replace Wg emanating
from the midgut needed to pattern the renal tubules [12], and in pupae Wg spreading is
important for normal development of the adult gut [13]. These results, combined with the
direct visualization of Wg protein as far away as 11 cells from its source in the wing disc [14],
support the model that Wg is not limited to juxtacrine signaling but rather spreads extracellu-
larly to signal at a distance from its source.

We have previously reported that Wg spreads from source cells in the *Drosophila* germar-
ium, the most anterior region of the ovary [15]. Extracellular Wg protein forms a visible gradi-
ent in the germarium, emanating from its source in the cap cells at the anterior tip of the
germarium and spreading posteriorly over a distance of about 50 μm or 5 cell diameters to
reach the follicle stem cells (FSCs) [16,17]; this extracellular spreading of Wg is facilitated by
the glypican Dally-like protein (Dlp), in a mechanism similar to that reported for the wing disc
[15,18,19]. In the germarium, Wg acts as a proliferative signal for the FSCs, inducing them to
increase their proliferation rate [15,17]. The progeny of FSCs, the follicle cells, encapsulate
cysts of germline cells to form immature *Drosophila* eggs, called egg chambers. Increasing the
level of Wg protein that reaches the FSCs, either by overexpressing *wg* in cap cells or by pro-
moting its extracellular spread, results in increased follicle cell numbers; in contrast, decreas-
ing the level of Wg protein that reaches the FSCs decreases the number of follicle cells,

resulting in encapsulation defects [15,17]. Because *wg in situ* hybridization signal and transcriptional reporters localize to cap cells, and because the visible Wg protein gradient is most concentrated at the cap cells, we concluded that Wg spreads in the germarium [15–17,20]. However, it has also been proposed that Wg signals in a juxtacrine manner to the FSCs, with the ligand emanating from neighboring escort cells, raising questions about the role of Dlp in this process [21,22]

The existence of the *NRT-wg* allele affords an opportunity to test the model that Wg spreads from the cap cells to the FSCs. If Wg signaling is extracellular over a distance, then NRT-Wg would not be able to substitute for wild-type Wg in the germarium; in contrast, if Wg signaling is juxtacrine in nature, emanating either from neighboring cells or presented via cytonemes from distant cells, then NRT-Wg would be able to substitute for wild-type Wg. In this study, we analyze the consequences of eliminating Wg spreading by tethering Wg in the germarium. In the course of these studies, we determined that NRT-Wg has an inherent dose-dependent toxicity, which we minimized by analyzing *NRT-wg/null* animals. We find that tethered Wg accumulates around the cap cells and cannot substitute for wild-type Wg in egg development. Further, FSC proliferation is reduced to undetectable levels when Wg is tethered, supporting the idea that wild-type Wg spreads in the germarium and has long-range function. Thus, Wg signals at a distance from the source cells in the germarium, a result that adds to the evidence that Wg signals at a distance and not only in a juxtacrine manner.

## Results and discussion

### Tethering Wg to the plasma membrane generates a dose-dependent toxicity

Previously it was reported that flies expressing membrane-tethered *NRT-wg* from the endogenous *wg* locus–flies which lack all wild-type Wg protein–have reduced fitness [10]. We quantified their survival to adulthood by crossing heterozygous balanced *NRT-wg* flies and counting the progeny classes (details of the *NRT-wg* and other genotypes used in this study are shown in Fig 1A). Full survival would be indicated by 33% homozygous *NRT-wg* flies because the *CyO* balancer is homozygous lethal; however, we found that only 8% of the progeny were homozygous *NRT-wg* (Fig 1B). The *NRT-wg* flies were generated by inserting a cDNA encoding *NRT-wg* into the *wg* locus, which was previously mutated to delete *wg* coding sequence. The appropriate controls for these *NRT-wg* flies have a wild-type *wg* cDNA inserted into the same deleted *wg* locus, which results in a homozygous viable, completely healthy line (called "control" from here on; see Fig 1A for schematic) [10]. In comparison to the homozygous *NRT-wg* flies, control flies survived to 5 days, as 32% of the progeny were homozygous control, remarkably close to the expected 33% (Fig 1B). If *NRT-wg* simply reduced *wg* function (e.g., by reducing Wg spreading) we would expect survival to decrease further when the dose of *NRT-wg* was halved, *in trans* to a null allele of *wg*. We crossed *NRT-wg* flies to *wg* null flies ($wg^{CX4}$) and counted *NRT-wg/null* flies and were surprised to discover that survival ratios were significantly improved, with *NRT-wg/null* flies representing 18% of the progeny (Fig 1B). These results indicate that *NRT-wg* is not simply a loss-of-function of Wg spreading.

The increased lethality of homozygous *NRT-wg* flies compared to *NRT-wg/null* flies can be explained either by a second mutation on the chromosome that contributes to the sickly phenotype when homozygous, or by *NRT-wg* itself having a weak neomorphic toxicity. To distinguish these possibilities, we utilized another allele, also engineered by the Vincent lab, which can be converted from wild-type *wg* to *NRT-wg* by FLP-mediated recombination (Fig 1A), without changing any other locus on the chromosome. We converted this allele to *NRT-wg* in the male germline with *TubP-FLP*; this newly flipped *NRT-wg* allele and the initial unflipped

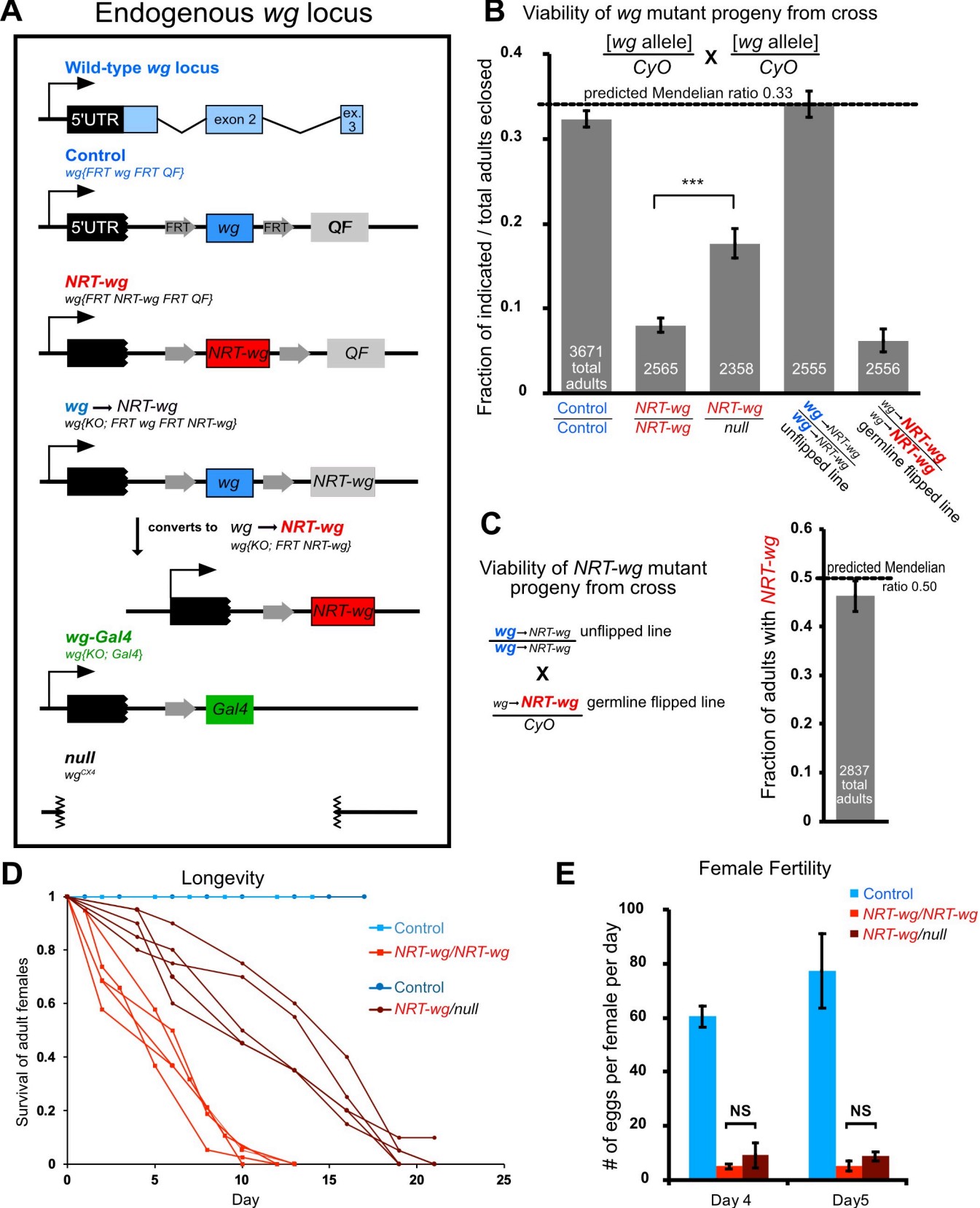

**Fig 1. Tethering Wg to the plasma membrane specifically reduced female fertility. A.** *wg* alleles used in this study. **B.** The *NRT-wg* allele has a dose-dependent toxicity inherent in the allele, not caused by a second chromosomal lesion. Homozygous *NRT-wg* flies (panel B, column 2) had severely compromised survival to adulthood, with homozygotes present at only 8% of total progeny rather than the predicted 33%. Survival was greatly improved (18% of total progeny) when flies carried one copy of *NRT-wg in trans* to the *wg^{CX4}* null allele (column 3, *wg-NRT/null)*. The FRT-containing (flippable) line was fully viable (34% of progeny) in its unflipped state, when it expressed wild-type *wg* (column 4). After germline excision of wild-type *wg*, *NRT-wg* was expressed in the same chromosomal background, and survival was severely compromised in homozygotes (6% of progeny, column 5). All parents carried the *CyO* balancer chromosome to standardize results, even though the control and unflipped line are homozygous viable and fertile. **C.** Heterozygous flies carrying the unflipped *wg* chromosome *in trans* to the flipped *NRT-wg* chromosome were fully viable (panel C, 46% of the population compared to 50% expected). **D.** *Homozygous NRT-wg* flies had severely reduced lifespan, but lifespan was improved in *wg-NRT/null* hemizygous flies. Each line represents an independent biological replicate of 16–38 flies. **E.** Both homozygous *NRT-wg* flies and hemizygous *NRT-wg/null* flies had dramatically reduced female fertility, indicating that fertility is specifically sensitive to Wingless tethering independent of its toxicity. Each bar represents 3 biological replicates of 5–12 females. Error bars represent SEM.

*wg* allele had identical chromosomal backgrounds. When we tested viability, we found that homozygotes of the unflipped *wg* line were completely viable whereas homozygotes of the newly flipped *NRT-wg* allele were barely viable (6% of the population, Fig 1B), mirroring our earlier results despite the identical background. As a final test of possible background elements contributing to lethality, we crossed the newly flipped *NRT-wg* line to the unflipped *wg* line, testing the viability of offspring heterozygous for *NRT-wg* yet homozygous for potential background elements. These progeny were viable (Fig 1C). Thus, despite extensive testing, we found no evidence to support the hypothesis that *NRT-wg* chromosomes carry a background mutation contributing to lethality, and we conclude NRT-Wg itself has a dose-dependent toxicity. Such toxicity could be explained by increased juxtacrine Wg signaling in neighbors of *wg*-expressing cells, since Wg that should spread to distant cells is tethered at the cell surface.

After eclosion, homozygous adult flies were difficult to work with because they were short-lived: even when fed well and housed without crowding, 50% of the homozygous *NRT-wg* female flies died by ~5 days after eclosion (Fig 1D). Like the eclosion phenotype, the lifespan phenotype was partially suppressed in *NRT-wg/null* flies: 50% lethality was observed by ~12 days after eclosion (Fig 1D). Thus, the dose-dependent toxicity of NRT-Wg was apparent in lifespan as well as in viability to adulthood.

## Tethered Wg specifically reduces female fertility

To determine whether Wingless tethering affected egg formation in the germarium, we examined the fertility of *NRT-wg* flies, mating them to wild-type males and counting the number of eggs laid per day. On days 4 and 5 after eclosion, control flies laid on average 61 and 78 eggs, whereas *NRT-wg* homozygous flies laid on average 5 eggs on each of these days, a reduction in fertility of 92–94% (Fig 1E). The sharp reduction in fertility indicates that Wg spreading is required for normal egg development. Importantly, with respect to fertility, hemizygous *NRT-wg/null* flies were not significantly different from homozygotes, with fertility reduced by 85–89% (Fig 1E), indicating that the *NRT-wg* fertility phenotype is not a result of the unexpected toxicity of homozygous *NRT-wg* and instead can be attributed to the lack of Wg spreading. These results are consistent with the model that Wg spreading is required for normal egg production.

Upon dissection, the ovaries of *NRT-wg* homozygous flies were observed to be much smaller than control ovaries with a reduction in the stages of vitellogenesis. In contrast, ovaries of *NRT-wg/null* hemizygotes were normally sized with representation of all stages of egg chamber development (Fig 2A–2D), suggesting that defects in later-stage vitellogenic eggs are caused by NRT-Wg toxicity, like the viability and lifespan phenotypes. Consistent with the relatively normal appearance of the ovaries, the architecture of the germarium of *NRT-wg/null* females appeared similar to control females, with egg chambers budded at the posterior of the germarium surrounded by a full complement of follicle cells (Fig 2E). The normal germarium

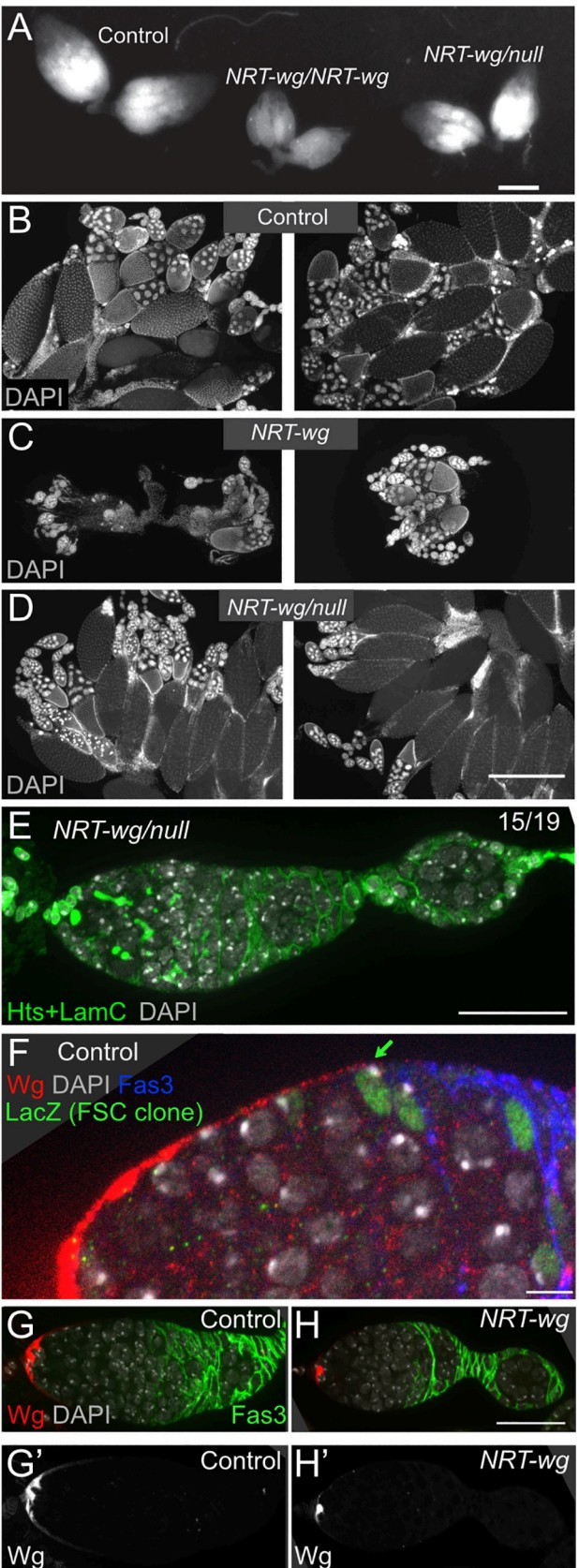

**Fig 2. Ovaries with tethered Wg have normal germarium architecture even though Wg protein is sequestered near cap cells. A.** Bright-field image showing sizes of control ovaries, *NRT-wg* homozygous ovaries, and *NRT-wg/null* hemizygous ovaries. *NRT-wg/null* hemizygotes are more similar than homozygotes to controls. Bar: 500 μm. **B-D.** DAPI staining in ovaries, 2 samples shown for each genotype, revealed the distribution of egg chamber stages in controls (B), in *NRT-wg* homozygous ovaries with fewer late-stage egg chambers (C), and in *NRT-wg/null* hemizygotes with a more typical distribution of egg chamber stages (D). Bar: 500 μm. All ovaries in A-D are from 6-day old females. **E.** Hemizygous *NRT-wg/null* germarium stained for DAPI (in blue) and Hts and LamC (in green) showing normal cap cells, spectrosomes/fusomes, and follicle cells. Like most *NRT-wg/null* germaria (15/19), this germarium does not show an encapsulation defect. Bar: 25 μm. **F.** Germarium, wild-type for *wg*, showing extracellular Wg protein (red) spreading from the anterior (left) to the follicle stem cells. A follicle stem cell (FSC) clone was genetically labeled to express *lacZ* (green). Bar: 5 μm. **G-H.** Control germarium (G) and *NRT-wg* homozygous germarium (H) stained for extracellular Wg protein (red) revealed that tethering Wg to its source in cap cells reduced the visible spread of Wg protein. Bar: 25 μm.

architecture was surprising, because previous studies by us and others have found that conditional loss of *wg* in adult females results in defects in the germarium, with too many germline cysts and not enough follicle cells to encapsulate them [15,17]. This mismatch of germline cysts to follicle cells can result in the accumulation of germline cysts in an enlarged germarium and/or encapsulation defects in which two germline cysts are squeezed together in one follicle cell covering [15,17].

In wild-type germaria, the Wg protein forms a gradient that extends from the anterior to the FSCs, identified by lineage tracing (Fig 2F) [15]. To validate the *NRT-wg* allele was tethered as expected, we examined Wg localization in *NRT-wg* homozygotes and in controls (with the same copy number of the *wg* gene). In contrast to the gradient of Wg protein found in the controls (Fig 2G), in *NRT-wg* germaria the Wg protein was sequestered at the cap cells, the cells that produce the Wg protein (Fig 2H). The change in Wg localization provided confirmation that the NRT transmembrane domain functioned in the germarium to tether Wg to the plasma membrane, but it did not explain why germarium architecture appeared normal when egg laying was so infrequent in *NRT-wg/null* females.

### *fz3-RFP* is a complex reporter of signaling by many Wnt ligands

Previously we reported *fz3-RFP* was a faithful reporter of Wnt signaling activity in the germarium [15]. The specificity of the *fz3-RFP* reporter for Wg has not been established, as four *Wnts* are expressed in the germarium (*Wnt2*, *Wnt4*, *Wnt6*, and *wg*; [16,20,23,24]). To determine if *fz3-RFP* responded to multiple Wnt ligands, we analyzed how *fz3-RFP* responded to the loss of each Wnt ligand (S1 Fig). We measured RFP fluorescence after knocking down each Wnt with a Gal4 driver in the cell type that generates it. *bab1*[Agal4-5] was used to knock down *wg* or *Wnt6* in cap cells (S1A–S1D Fig), and *C587-Gal4* was used to knock down *Wnt2* or *Wnt4* in escort cells (S1E–S1J Fig). These experiments showed that *fz3-RFP* reports the signaling activities of at least three Wnt ligands–Wg, Wnt2, and Wnt4: when *wg* was knocked down in cap cells with *bab1*[Agal4-5], RFP fluorescence decreased by 35% compared to controls; when *Wnt2* was knocked down in escort cells with *C587-Gal4* driving either of two RNAi lines, RFP fluorescence decreased by 51 or 45%; when *Wnt4* was knocked down in escort cells, RFP fluorescence decreased by 69%. In contrast RFP levels were not affected by knocking down *wg* in escort cells, confirming that *wg* is not expressed in escort cells at appreciable levels (S1J Fig). To analyze *fz3-RFP* in *NRT-wg* germaria, we needed to recombine these two loci on the 2nd chromosome. *fz3-RFP* is unmapped and identifying a recombinant required molecular screening of hundreds of progeny, suggesting that it is tightly linked to the *wg* locus. Puzzlingly, when germaria were analyzed from the *fz3-RFP*, *NRT-wg* recombinant *in trans* to a *wg* null allele (*fz3-RFP*, *NRT-wg/null*), RFP levels were unexpectedly increased throughout the anterior germarium (S2 Fig). Although *fz3-RFP* reports the activity of three Wnt ligands, these results

suggest there may be a relay of Wnt signaling from the cap cells to the escort cells, and proper levels depend on Wg spreading. We conclude that *fz3-RFP* is a complex reporter for many types of Wnt signaling, and that the Wnt signaling pattern in the germarium indicated by this reporter is the sum of signaling from several ligands.

## Wg spreading is required for FSC proliferation

Previously we found that FSC proliferation is sensitive to the levels of Wg spreading from the cap cells, with more proliferation occurring when the range of Wg spreading is visibly increased. Because the range of Wg spreading is sharply limited by NRT-Wg, we hypothesized that FSC proliferation would be reduced, a possible explanation for the lack of fertility in *NRT-wg* females. In our earlier study, FSC proliferation was measured by two assays–indirectly by the frequency of FSC mitotic clone induction (see Fig 2F), and directly by visualizing mitotic cells in the region of FSCs by phospho-histone H3 (pH3) staining–and the results of these two assays were in close agreement [15]. For this study, we were unable to use lineage tracing to evaluate FSC proliferation in the *NRT-wg/null* germaria we were analyzing, as the allele we used contains an *FRT* at the *wg* locus, precluding the use of *FRT*s for mitotic recombination. Fortunately, we were able to use the alternative approach of direct visualization of mitotic cells with anti-pH3 (Fig 3A and 3B), and we analyzed their frequency in the germarium. Currently it is thought that there are three layers or rings of FSCs, extending anteriorly from the 2a/2b boundary, with proliferation occurring mostly in the posterior ring at the boundary of bright Fas3 staining [25–27]. This understanding of FSCs is at odds with an older model, which held there were only two FSCs [28,29], but even in this model, FSC proliferation is at the border between regions 2a and 2b, at the boundary of bright Fas3 antibody staining [29–32]. Thus, we counted pH3 stained nuclei at this border. For comparison, we also measured mitosis in other regions of the germarium: the germline cells found anterior to the 2a/2b boundary; region 2b where mitotic cells are prefollicle cells, the immediate daughters of the FSCs; region 3 where mitotic cells are follicle cells; and stage 6, an ovarian tissue outside the germarium. We analyzed three sets of *NRT-w*g genotypes. As described previously, we analyzed germaria from homozygous *NRT-wg* females (Fig 3C) and from hemizygous *NRT-wg/null* females (Fig 3D), as toxicity is reduced in the latter. We also analyzed germaria from females carrying a conditional allele, in which *wg* was converted to *NRT-wg* only after adults eclosed, so as to avoid any contribution of *NRT-wg* during pupal development of the ovary; the final adult genotypes compared were *control-wg* (unflipped)/ *null* versus *NRT-wg* (flipped)/*null* (Fig 3E). Examining all three data sets, it is striking that pH3 staining was never observed in the FSC-containing region in any of the 156 *NRT-wg* germaria, compared to 33 mitotic pH3-positive cells observed in 131 control ovaries. Although we counted only the pH3-positive cells, DAPI staining showed the total number of cells to be similar, consistent with our observations that the germarium is not disrupted in these mutants (Fig 2D and 2E). In the other regions, some reductions in mitosis were observed at varying levels in different *NRT-wg* backgrounds, with the reduction in mitosis most pronounced in the homozygous *NRT-wg*, but they were not consistent across genotypes. Together, these results indicate that in *NRT-wg* germaria, the FSC proliferation rate is so low as to be undetectable, and such a low rate would explain the ten-fold reduction in egg-laying. These results are consistent with the model that Wg spreads from a distance to signal the FSCs, a process previously shown to be mediated by the glypican Dlp [15].

## Several mechanisms compensate for the reduction in FSC proliferation

Given the absence of measurable FSC proliferation, we were puzzled by the architecture of the *NRT-wg* germaria, since germline proliferation was not as dramatically reduced as follicle stem

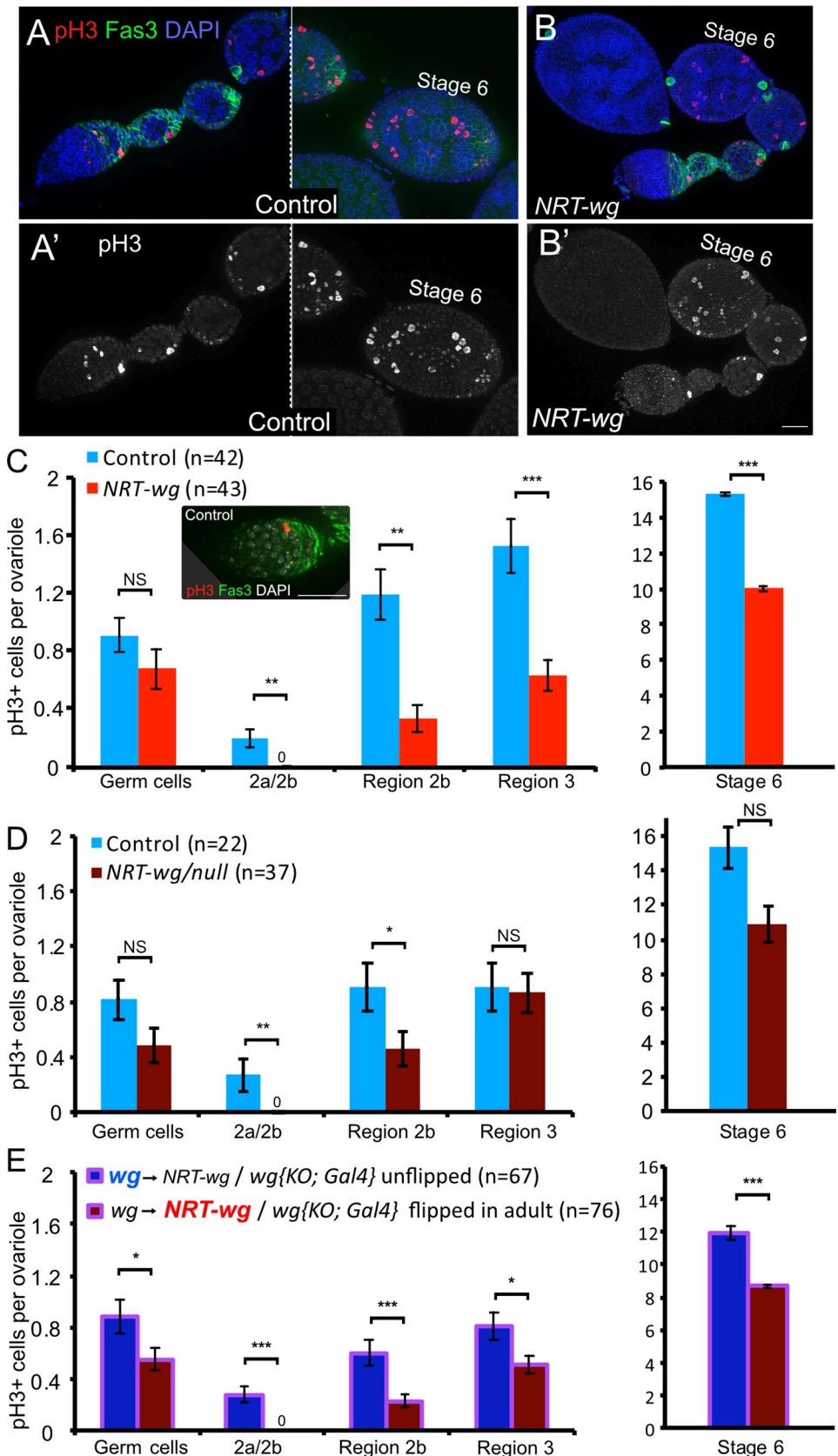

**Fig 3. Follicle stem cell proliferation was severely reduced when Wg was tethered to the plasma membrane. A-B**. Cell proliferation was detected by anti-phospho-H3 staining, which recognizes mitotic cells in germaria and early egg chambers. Scale bar: 25 μm. **C**. In homozygous *NRT-wg* ovaries, significantly less cell proliferation occurred in all somatic cell types examined, although germ cell proliferation was not significantly changed. Inset shows a FSC at the 2a/2b boundary labeled by anti-pH3 in a control germarium, scale bar 25μm. **D**. In hemizygous *NRT-wg/null* ovaries, decreases in somatic cell proliferation were observed specifically in the region of follicle stem cells (Fas3- region) and their immediate daughters (Region 2b), and not in later Regions 3 or 6. **E**. In germaria that were raised with control Wg then flipped to *NRT-wg* as adults, decreased cell proliferation occurred, with the most pronounced differences in the region of follicle stem cells (Fas3- region) and their immediate daughters (Region 2b). * $p < 0.05$, ** $p < 0.01$, *** $p < 0.001$, NS not significant. Error bars are SEM.

cell proliferation (Fig 3C–3E). Such a mismatch in proliferation rates would result in either the accumulation of too many germline cells in the germarium, encapsulation defects with multiple germline cysts within one follicle, and/or death of germline cells. We noted that some *NRT-wg/ null* germaria displayed encapsulation defects (15/19 germaria), although this defect was not sufficiently penetrant to compensate on its own for the drastic reduction in FSC proliferation. To assay germline death, we performed TUNEL-staining to identify apoptotic cells in germaria (Fig 4A–4C). In *NRT-wg/null* germaria, the rate of germline death was slightly but significantly elevated over controls, which combined with the encapsulation defects may compensate for the lack in FSC proliferation. Remarkably, in *NRT-wg* homozygotes nearly every germarium contained a dying germline cyst. These two mechanisms together, apoptosis of unencapsulated cysts and encapsulating multiple cysts into one follicle, likely compensate for the continued proliferation of germline stem cells while the follicle stem cells do not proliferate.

It has been documented that homozygous *NRT-wg* animals have defects in gut development during pupal metamorphosis, specifically having defects in muscle patterning, cell fate specification, organ folding, and the relative size of various regions [13]. It is reasonable to consider that *NRT-wg* malformed guts may result in reduced nutritional uptake in adults, and interestingly, several of the ovarian phenotypes we observed in *NRT-wg* homozygotes—small ovaries, a reduction in follicle cell proliferation, and apoptosis of germline cysts—are observed in wild-type flies with poor nutrition [32,33]. This nutrition interpretation is less likely to explain the reduced fertility of *NRT-wg/null* hemizygotes because ovary size was normal and overall follicle cell proliferation was not significantly different from controls except for the region of follicle stem cells and their immediate daughters, but nevertheless it is impossible to rule out developmental causes when Wg was tethered throughout development. To separate the role of Wg spreading in development from oogenesis, we generated conditional mutants in which Wg was tethered only after eclosion, using {*FRT wg FRT NRT-wg*} which converted *wg* to *NRT-wg* in adults via intrachromosomal recombination (Fig 4D and 4E). To convert this allele, we first expressed the FLP recombinase ubiquitously with *Act5C-Gal4* restricted by Gal80^ts to be expressed only after eclosion, but few adults were recovered. Fortunately, expressing FLP recombinase in the *wg* expression pattern with *wg-Gal4*, restricted by Gal80^ts to adults, resulted in *NRT-wg/null* viable adults. Although these germaria did not generally display encapsulation defects (only 1/32 had a visible encapsulation defect), they did display a significant increase in 16-cell germline cysts (Fig 4F–4H), a previously reported phenotype for the adult-onset loss of *wg* [17]. These results are consistent with the model that extracellular spreading of Wg is required in the germarium for normal FSC proliferation, and without sufficient FSC proliferation, the germarium backs up with too many germline cysts.

## Extracellular spreading of Wg may be part of a complex relay

In these *NRT-wg* experiments, and in previously reported *wg* loss-of-function germarium experiments, the *wg* gene product was manipulated throughout the animal [17]. Thus, these

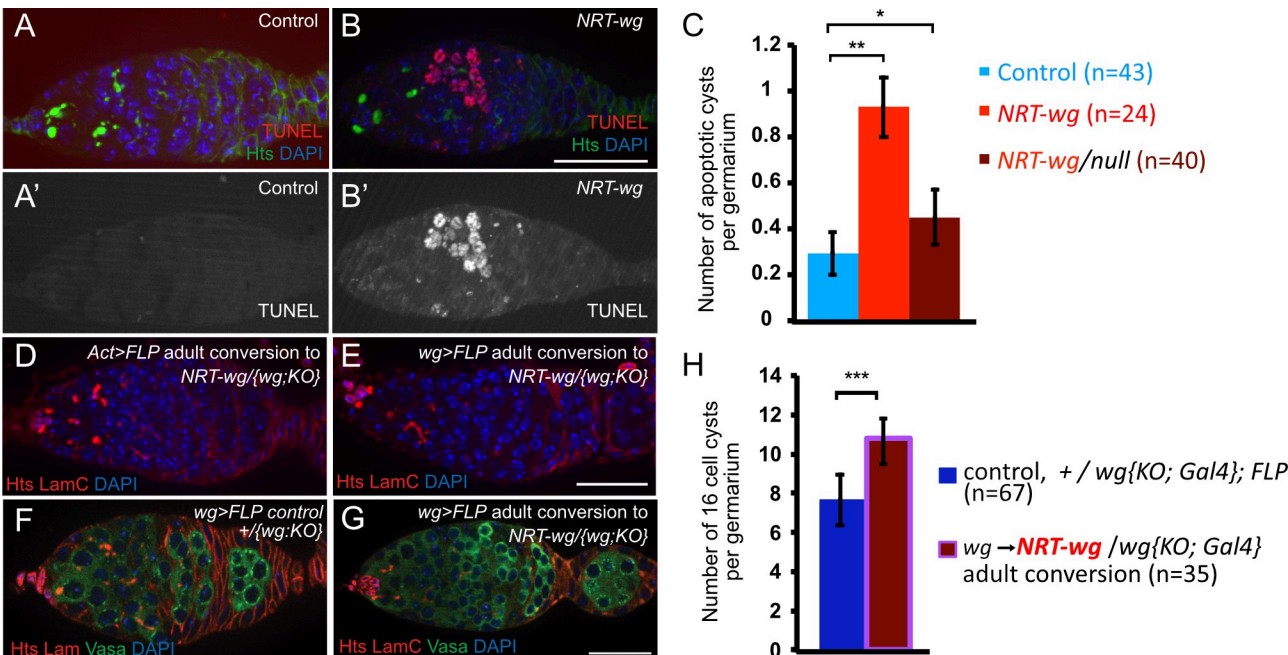

**Fig 4. Germaria with tethered Wingless had too many germline cysts. A–C.** Apoptotic cysts were detected in germaria by TUNEL. Controls expressing wild-type *wg* (A) showed few apoptotic cysts, whereas germaria from flies expressing only *NRT-wg* throughout development had substantially more apoptotic germline cysts (B), on average about 1 apoptotic cyst per germarium (C). Bar: 25 μm. **D–E.** The *wg* locus was converted to express *NRT-wg* during adulthood by either *Act>FLP* (D) or *wg>FLP* (E) under *Gal80^{ts}* control. Egg encapsulation proceeded normally in nearly all germaria, 35/40 germaria for *Act>FLP* and 31/32 for *wg>FLP*. Bar: 25 μm. **F–H.** When the *wg* locus is converted to *NRT-wg* during adulthood by *wg>FLP*, germaria contain increased numbers of germline cysts (G) compared to controls in which no conversion occurred (F). Germline cysts are evident with anti-Vasa staining (green). Bar: 25 μm. Error bars are standard deviation.

experiments cannot completely exclude the possibility that *wg* function and Wg spreading are required in another adult tissue to promote FSC proliferation. A conclusive experiment about whether Wg spreads from the cap cells to the FSCs would be to convert *wg* to *NRT-wg* only in the cap cells (converting *wg* to a null allele as a control). Unfortunately, despite substantial efforts we have not been unable to identify a cap-cell specific Gal4 driver that expresses in all of the cap cells, the source of most or all Wg for the germarium, and the escaping cap cells appear to provide sufficient Wg protein to maintain signaling. Nevertheless, because NRT-Wg is capable of signaling to neighboring cells [13], these data clearly demonstrate that extracellular Wg spreading is required for oogenesis and follicle stem cell proliferation. Because Wingless spreading is not required in the wing for patterning but is required for egg production, our findings suggest that different tissues may have different requirements for diffusible versus juxtacrine wingless signaling.

On its own, our data are consistent with a simple model that Wg spreads from the cap cells to the FSCs where it directly signals FSC proliferation. We developed this model from our previous study, where we found that the glypican Dlp plays a central role in spreading Wg to the FSCs, and those data are consistent with our results here. This simple model is challenged, however, by studies of FSCs that cannot transduce Wnt signals. In two papers, the Kalderon lab has found that when FSC clones decrease Wnt signaling by overexpressing *dnTCF* [25] or when they lose Wnt signaling by mitotic recombination of an *arrow* mutant [27], FSCs have a small decrease in proliferation measured by EdU incorporation, a decrease insufficient to explain our data that proliferation rates are below the threshold of detection when Wg is tethered. Interestingly, the Kalderon group found that increasing activity of the Wnt signaling

pathway dramatically reduces FSC proliferation [25]. One model to reconcile these conflicting data is that Wg indirectly signals FSC proliferation, through an intermediate cell type that cannot be reached with NRT-Wg. Our *fz3-RFP* Wnt reporter data suggest that total Wnt signaling is unexpectedly increased when Wg is tethered, hinting at a complex relay of Wnt signaling in the germarium. Thus the loss of FSC proliferation in *NRT-wg* germaria may be indirectly caused by the increase in Wnt signaling. Together, these data suggest that Wg spreading is part of a complex system of Wnt regulation required for FSC proliferation.

## Methods

### Fly stocks

Flies were cultured on cornmeal-molasses media at 25˚C unless otherwise noted. Females were fed on media with wet yeast paste in the presence of $w^{1118}$ males, and tossed to new vials every 1–2 days until dissection. The following stocks were a generous gift from Dr. Jean-Paul Vincent: *wg{KO; FRT NRT-wg FRT QF}/CyO. wg{KO; FRT wg FRT QF}/CyO. wg{KO; Gal4/ CyO}. wg{KO; FRT wg FRT NRT-wg}/CyO. wg{KO; Gal4]*. The *NRT-wg* flies used in this study were *wg{KO; FRT NRT-wg FRT QF}*, which express *NRT-wg* and not *QF* in their unflipped state; the corresponding control flies were *wg{KO; FRT wg FRT QF}*, which express *wg* and not QF in their unflipped state. We did not use the non-flippable *wg{KO; NRT-wg}* line because of reported problems with its control, *wg{KO; wg}*. In its unflipped state, the *wg{KO; FRT NRT-wg FRT QF}* chromosome is equivalent to *wg{KO; NRT-wg}*. The $wg^{CX4}$ null allele was obtained from the Bloomington *Drosophila* Stock Center (BDSC).

For adult-onset tethering of Wg in which *wg* is replaced with *NRT-wg*, the cross *wg{KO; Gal4}/CyO; Tub-Gal80^{ts}/TM6 x wg{KO; FRT wg FRT NRT-wg}/CyO; UAS-FLP/TM6* was performed at 18˚C. Immediately after progeny eclosed as adults, females were collected of genotype *wg{KO; Gal4}/wg{KO; FRT wg FRT NRT-wg}; UAS-FLP/Tub-Gal80^{ts}* and moved to 29˚C for 5 days (Fig 3E) or 10 days (Fig 4) before dissection. *UAS-FLP*, *Tub-Gal80^{ts}*, and the alternative driver *Act5C-Gal4* were obtained from the BDSC.

FSC mitotic clones (Fig 2F) were induced and stained for lacZ as previously described [15]

To assess *fz3-RFP* expression in control and Wnt knockdown conditions, *fz3-RFP; bab1^{A-gal4-5}, Tub-Gal80^{ts}* or *C587-Gal4; fz3-RFP; Tub-Gal80^{ts}* females were crossed to $w^{1118}$ (control), *UAS-wg^{RNAi}* (NIG-FLY 4889 R-4), *UAS-Wnt2^{RNAi}* (BL 29441, BL 36722), *UAS-Wnt4^{RNAi}* (BL 29442) or *UAS-Wnt6^{RNAi}* (VDRC 104020) males at 18˚C. F1 females of the appropriate genotypes were collected within one day of eclosing, crossed with $w^{1118}$ males, transferred to fresh vials supplemented with yeast paste and incubated at 29˚C for Gal80^{ts} inactivation and Gal4/ UAS-RNAi-mediated knockdown of Wnts for 7 days. During the incubation at 29˚C, flies were transferred to fresh vials supplemented with yeast paste every other day.

To assess *fz3-RFP* expression in germaria in control and *NRT-wg* heterozygous flies, *fz3-RFP/CyO* or *fz3-RFP, wg{KO; FRT NRT-wg FRT QF, Pax-Cherry}/CyO* flies were analyzed. To assess *fz3-RFP* expression in germaria of *NRT-wg* homozygotes, *fz3-RFP, wg{KO; FRT NRT-wg FRT QF, Pax-Cherry}/wg{KO; FRT NRT-wg FRT QF, Pax-Cherry}* were analyzed. To assess fz3-RFP in *NRT-wg/null flies*, *fz3-RFP, wg{KO; FRT NRT-wg FRT QF, Pax-Cherry}/ wg^{CX4}* were analyzed. All crosses were performed at 25˚C. Females of the appropriate genotypes were aged at 25˚C for 5–7 days.

## Genotypes in each figure panel

**Fig 1B:**

 1^{st} column: *wg{FRT wg FRT QF}/wg{FRT wg FRT QF}*
 2^{nd} column: *wg{FRT NRT-wg FRT QF}/wg{FRT NRT-wg FRT* QF}

3rd column: *wg{FRT NRT-wg FRT QF}/wg^{CX4}*

4th column: *wg{KO; FRT wg FRT NRT-wg}/ wg{KO; FRT wg FRT NRT-wg}*

5th column: *wg{KO; FRT NRT-wg}/wg{KO; FRT NRT-wg}* (germline *NRT-wg* line generated by flipping out *FRT wg FRT* cassette in the germline of *wg{KO; FRT wg FRT NRT-wg}/wg{KO; FRT wg FRT NRT-wg}* with *tub-FLP*)

**Fig 1C:** numerator of fraction: *wg{KO; FRT wg FRT NRT-wg}/wg{KO; FRT NRT-wg}*

Denominator is sum of two genotypes: *wg{KO; FRT NRT-wg}/ wg{KO; FRT wg FRT NRT-wg} + wg{KO; FRT wg FRT NRT-wg}/CyO*

**Fig 1D:**

Both controls: *wg{FRT wg FRT QF}/wg{FRT wg FRT QF}*

NRT-wg/NRT-wg: wg{FRT NRT-wg FRT QF}/wg{FRT NRT-wg FRT QF}

NRT-wg/null: wg{FRT NRT-wg FRT QF}/wg^{CX4}

**Fig 1E:**

Control: *wg{FRT wg FRT QF}/wg{FRT wg FRT QF}*

*NRT-wg/NRT-wg*: *wg{FRT NRT-wg FRT QF}/wg{FRT NRT-wg FRT QF}*

*NRT-wg/null*: *wg{FRT NRT-wg FRT QF}/wg^{CX4}*

**Fig 2:**

A left, B, and G: *wg{FRT wg FRT QF}/wg{FRT wg FRT QF}*

A middle, C, and H: *wg{FRT NRT-wg FRT QF}/wg{FRT NRT-wg FRT QF}*

A right, D, and E: *wg{FRT NRT-wg FRT QF}/wg^{CX4}*

F: *hsFLP/+; X15-33/X15-29*

**Fig 3:**

A, C control, D control: *wg{FRT wg FRT QF}/wg{FRT wg FRT QF}*

B, C: *wg{FRT NRT-wg FRT QF}/wg{FRT NRT-wg FRT QF}*

D: *wg{FRT NRT-wg FRT QF}/wg^{CX4}*

E: Control: *wg{KO; FRT wg FRT NRT-wg}/wg{KO; Gal4}*

Flipped as adults: *wg{KO; FRT wg FRT NRT-wg}/wg{KO; Gal4}; UAS-FLP/Tub-Gal80^{ts}*

**Fig 4:**

A: *wg{FRT wg FRT QF}/wg{FRT wg FRT QF}*

B: *wg{FRT NRT-wg FRT QF}/wg{FRT NRT-wg FRT QF}*

C: *wg{FRT wg FRT QF}/wg{FRT wg FRT QF}*

wg{FRT NRT-wg FRT QF}/wg{FRT NRT-wg FRT QF}

wg{FRT NRT-wg FRT QF}/wg^{CX4}

D: *wg{KO}/wg{KO; FRT wg FRT NRT-wg}; Actin5C-Gal4, Tub-Gal80^{ts}/UAS-FLP*

E: *wg{KO; Gal4}/wg{KO; FRT wg FRT NRT-wg}; Tub-Gal80^{ts}/UAS-FLP*

F: *wg{KO; Gal4}/+; Tub-Gal80^{ts}/UAS-FLP*

G: *wg{KO; Gal4}/wg{KO; FRT wg FRT NRT-wg}; Tub-Gal80^{ts}/UAS-FLP*

H: control: *+/wg{KO; Gal4}; UAS-FLP/Tub-Gal80^{ts}*

Flipped as adults: *wg{KO; Gal4}/wg{KO; FRT wg FRT NRT-wg}; UAS-FLP/Tub-Gal80^{ts}*

**S1 Fig:**

A: *w^-/w^-; fz3-RFP/+; bab1^{Agal4-5}, Tub-Gal80^{ts}/+*

B: *w^-/w^-; fz3-RFP/UAS-wg^{RNAi}; bab1^{Agal4-5}, Tub-Gal80^{ts}/+*

C: *w^-/y^-,w^-; fz3-RFP/UAS-Wnt6^{RNAi}; bab1^{Agal4-5}, Tub-Gal80^{ts}/+*

D: Genotypes A-C are shown on the graph

E: *C587-Gal4/w^-; fz3-RFP/+; Tub-Gal80^{ts}/+*

F: *C587-Gal4/w^-; fz3-RFP/ UAS-wg^{RNAi}; Tub-Gal80^{ts}/+*

G: *C587-Gal4/y^-, v^-; fz3-RFP/ UAS-Wnt2^{RNAi}; Tub-Gal80^{ts}/+* (Line #1)

H: *C587-Gal4/ y^-, sc^-, v^-; fz3-RFP/ UAS-Wnt2^{RNAi}; Tub-Gal80^{ts}/+* (Line # 2)

I: *C587-Gal4/ y⁻, v⁻; fz3-RFP/+; Tub-Gal80^{ts}/ UAS-Wnt4^{RNAi}*

J: Genotypes E-I are shown on the graph

**S2 Fig:**

A: *w⁻/w⁻; fz3-RFP/CyO; +/+*

B: *w⁻/w⁻; fz3-RFP, wg{KO; FRT NRT-wg FRT QF, Pax-Cherry}/CyO*

C: *w⁻/w⁻; fz3-RFP, wg{KO; FRT NRT-wg FRT QF, Pax-Cherry}/wg{KO; FRT NRT-wg FRT QF, Pax-Cherry}*

D: *w⁻/w⁻; fz3-RFP, wg{KO; FRT NRT-wg FRT QF, Pax-Cherry}/wg^{CX4}*

E: Genotypes A-D are shown on the graph

## Tissue fixation and staining

Ovaries were stained as previously described [15]. Briefly, ovaries were dissected in Schneider's *Drosophila* media, fixed in 4% paraformaldehyde (Ted Pella) in PBS and washed with PBST (PBS + 0.1% Triton X-100) before being blocked with 5% normal goat serum in PBST and incubated with primary antibodies at 4°C overnight. To visualize DNA, ovaries were incubated in 1 μg/ml DAPI (Sigma) in PBST for 10 min. Extracellular Wg staining was performed according a published protocol [6] and was also described in [15]. The following primary antibodies were from Developmental Studies Hybridoma Bank (DSHB): mouse anti-Fas3 (7G10, 1:8), mouse anti-Hts (1B1, 1:5), mouse anti-LamC (LC28.26, 1:20), mouse anti-Wg (4D4, 1:3) and rat anti-Vasa (1:10). Rabbit anti-phospho-Histone H3 (Millipore, 1:1,000) was used to label mitotic cells. Secondary antibodies used were goat anti-rabbit IgG and goat anti-rat IgG conjugated to Alexa Fluor 488 (Molecular Probes), Cy3-conjugated goat anti-mouse IgG1 or IgG2a. Stained samples were mounted in Vectashield (Vector Laboratories).

For Fz3-RFP visualization, ovaries were dissected in incomplete Schneider's media, fixed in 4% paraformaldehyde (Ted Pella) in 1XPBS at RT, washed with 1X PBST, and mounted in Vectashield mounting media containing DAPI (Vector Laboratories).

For TUNEL staining, ovaries were dissected and fixed as described above, washed thoroughly in PBS, and permeabilized with PBS containing 0.1% Triton X-100 and 0.1% sodium citrate. 100 μl of TUNEL reaction mixture (*In Situ* Cell Death Detection Kit TMR Red, Roche) was added per 5 pairs of ovaries, and samples were incubated at 37°C in the dark for 1 hr. Ovaries were washed thoroughly in PBST, blocked and co-stained with primary antibodies overnight as described above.

## Microscopy and imaging

Fluorescent images were taken by a Zeiss Axioimager M2 equipped with an Apotome system and an AxioCam MRm camera (Zeiss). Images were acquired using 40X/1.3 oil EC Plan-Neo-Fluor or 63X/1.4 oil Plan-Apochromat objective lens at room temperature. The Zeiss Axiovision 4.8 software was used for data acquisition, and projections of Z-stacks were compiled using the Orthoview functions. Images were exported as 16-bit TIFF files and processed with Adobe Photoshop CS4 or Affinity Photo. Brightfield images of ovaries were acquired with a Zeiss Axiocam MRc camera mounted to a Zeiss Lumar V12 stereomicroscope, using a Neolumar S 1.5x objective, X-Cite 120Q light source and Axiovision 4.8 software.

## Quantification and statistics

Viability (eclosion) assays were performed at 25°C. Parents were allowed to lay eggs in bottles for 2–3 days. Adults emerged ~10 days later, and all progeny were counted until they stopped eclosing (never longer than 8 days). For the crosses in Fig 1B, 8 bottles (replicates) were counted for control and *NRT-wg* homozygotes; 4 bottles for *NRT-wg/null*; and 6 bottles each

for the germline flipped and non-flipped genotypes. For the cross in Fig 1C, 6 bottles were counted. For longevity assays, about 20 adult females were collected upon eclosion and cultured with $w^{1118}$ males in a vial. Dead females were counted every 2 days when flies were tossed to new vials.

For the fertility assay, female flies were fattened on cornmeal-molasses media with males and fresh yeast for 2–4 days, then 10–12 females and half as many males were put in an egg-collecting cage fed daily with a grape juice plate with yeast paste. Each 24 hours the plate was changed, eggs laid on the plate were counted, and the number of eggs/female/day was determined by dividing by the number of eggs by the number of females present at the start of the collection period. Dead females were removed daily and the number of surviving females was recorded for the next day's egg collection.

For quantification of *fz3-RFP* intensities, Z-stacks spanning the entire germarium were acquired at 20X for each genotype, sum Z projection of each germarium was generated in ImageJ, and average *fz3-RFP* intensity was obtained by outlining the *fz3-RFP* expression domain in the germarium. All germaria were imaged at same settings for comparisons shown in individual graphs.

To determine the number of mitotic cells in regions of the ovary, ovarioles were dissected and stained with anti-phospho-Histone H3 (pH3), Fas3, and DAPI. Mitotic cells were recognized by anti-pH3 staining. DAPI and Fas3 were used to identify the region of the ovary. Cells at the 2a/2b border, proximal to the region of high Fas3 staining, were considered as FSCs. Sixteen-cell cysts were identified by their typical structure of fusomes labeled with anti-Hts staining and vasa germline staining.

For main figures, student's *t*-test (two-tailed, two-sample equal variance) was used for statistical analysis and a *p* value of <0.05 was considered significant. For dot plots in S1D and S1J and S2E Figs, comparisons of means across indicated genotypes was done using ordinary one-way ANOVA, and significant differences between genotypes was determined by Tukey's test. Mean and standard error of mean are represented on the dot plots.

## Supporting information

**S1 Fig. *fz3-RFP* reports signaling activity of Wg, Wnt2, and Wnt4 in the *Drosophila* germarium. A-J.** The Wnt signaling reporter *fz3-RFP* in red, nuclei stained with DAPI in blue. Scale bar: 20 μm. **A-C.** *bab1*$^{Agal4-5}$, expressed in cap cells and terminal filament cells, was used to knock down *wg* (B) or *Wnt6* (C), both expressed in cap cells. **D.** Quantification of *fz3-RFP* intensity. Knockdown of *wg* resulted in decreased *fz3-RFP* expression, whereas knockdown of *Wnt6* did not affect *fz3-RFP* expression. **E-I.** *C587-Gal4*, strongly expressed in escort cells, was used to knockdown *wg* (F), *Wnt2* (G,H) or *Wnt4* (I). *Wnt2* and *Wnt4* are expressed in escort cells. **J.** Quantification of *fz-RFP* intensity. Levels are comparable in controls and when *wg* is knocked down in escort cells, consistent with previous findings that *wg* is expressed in cap cells, not escort cells. Knockdown of *Wnt2* in escort cells using two independent RNAi lines or knockdown of *Wnt4* in escort cells results in decreased *fz3-RFP* expression. ** indicates p value: 0.0099–0.001, *** indicates p value <0.00099, n.s. indicates not statistically significant. Each dot in D and J indicates average RFP intensity for a single germarium. n = 6–8 germaria per genotype. AU: Arbitrary Unit.
(PDF)

**S2 Fig. *fz3-RFP* expression is higher *NRT-wg* germaria. A-D.** A comparison of *fz3-RFP* expression in the germaria of (A) control (B) *NRT-wg* heterozygous, (C) *NRT-wg* homozygous, and (D) *NRT-wg*/*null* flies shows that membrane tethering of Wg unexpectedly induces *fz3-RFP* expression in germaria. DAPI labels nuclei. AU: Arbitrary units. Scale bar: 20 μm. **E.**

Quantification of *fz3*-RFP intensity in germaria of the indicated genotypes. Each dot indicates average *fz3-RFP* intensity in one germarium. n = 8–10 germaria per genotype. * indicates p value: 0.01–0.05, ** indicates p value: 0.0099–0.001, *** indicates p value < 0.00099. (PDF)

**S1 Table.** An excel file containing raw data for figure panels 1B-C, 1D, 1E, 3 C-D, and 4H. (XLSX)

## Acknowledgments

We thank Cyrille Alexandre and Jean-Paul Vincent as well as the Bloomington Drosophila Stock Center for fly stocks, the Developmental Studies Hybridoma Bank for antibodies.

## Author Contributions

**Conceptualization:** Xiaoxi Wang, Andrea Page-McCaw.

**Data curation:** Xiaoxi Wang, Kimberly S. LaFever, Andrea Page-McCaw.

**Formal analysis:** Xiaoxi Wang, Kimberly S. LaFever, Indrayani Waghmare, Andrea Page-McCaw.

**Funding acquisition:** Andrea Page-McCaw.

**Investigation:** Xiaoxi Wang, Kimberly S. LaFever, Indrayani Waghmare, Andrea Page-McCaw.

**Methodology:** Xiaoxi Wang, Indrayani Waghmare, Andrea Page-McCaw.

**Project administration:** Andrea Page-McCaw.

**Resources:** Andrea Page-McCaw.

**Supervision:** Andrea Page-McCaw.

**Validation:** Indrayani Waghmare, Andrea Page-McCaw.

**Visualization:** Xiaoxi Wang, Kimberly S. LaFever, Indrayani Waghmare.

**Writing – original draft:** Xiaoxi Wang, Andrea Page-McCaw.

**Writing – review & editing:** Indrayani Waghmare, Andrea Page-McCaw.

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
