## [Decision Letter · Decision Letter 0]

19 Jan 2021

Dear Dr Page-McCaw,

Thank you very much for submitting your Research Article entitled 'Extracellular spreading of Wingless is required for Drosophila oogenesis' to PLOS Genetics.

The manuscript was fully evaluated at the editorial level and by independent peer reviewers. The reviewers appreciated the attention to an important problem, but raised some substantial concerns about the current manuscript. Based on the reviews, we will not be able to accept this version of the manuscript, but we would be willing to review a much-revised version. We cannot, of course, promise publication at that time.

In reading through the detailed reviewer's comments, there are several additional experiments that are suggested, along with many questions/concerns about the interpretation of your results.  I don't think additional data are required for you to address the reviewer's points.  But I ask that you carefully consider the reviewer's concerns, and address them thoroughly in the rebuttal.  In addition, I ask you to modify the text of your manuscript where appropriate to take the reviewer's views into account.

One additional comment about the fz3-RFP reporter: I feel that both you and the reviewer give this reporter too much credit as an accurate measure of Wnt signaling.  RFP expression is driven by an endogenous stretch of regulatory DNA that may respond to inputs in addition to Wnt/beta-catenin signaling.  The reviewer feels the increase in fz3-RFP in NRT-wg backgrounds (including NRT-wg/CyO - do those flies have normal patterning and fertility?) suggests that NRT-wg/null is similar to Axin or Apc clones.  That seems unlikely and we don't have a direct comparison, i.e., fz3-RFP expression in those mutant backgrounds.  I agree with your view that the fz3-RFP reporter is complex, and this complexity may involve more than Wnt signaling (at least in a NRT-wg background).

If you decide to revise the manuscript for further consideration at PLOS Genetics, please aim to resubmit within the next 60 days, unless it will take extra time to address the concerns of the reviewers, in which case we would appreciate an expected resubmission date by email to plosgenetics@plos.org.

[LINK]

We are sorry that we cannot be more positive about your manuscript at this stage. Please do not hesitate to contact us if you have any concerns or questions.

Yours sincerely,

Ken M. Cadigan

Guest Editor

PLOS Genetics

Gregory P. Copenhaver

Editor-in-Chief

PLOS Genetics

Reviewer's Responses to Questions

**Comments to the Authors:**

Reviewer #1: This manuscript uses a membrane-tethered form of Wingless to address how spreading of an extracellular morphogen influences follicle stem cell proliferation in the fly ovary. This is a nice follow-up of earlier work from the same lab showing that Dally-like Protein is essential for follicle stem cells, presumably through its role in regulating extracellular Wingless. The results presented here not only establish the importance of Wingless protein spreading to the follicle stem cells, but also reveal problems with the original membrane-tethered Wingless construct. In addition, the authors have found interesting complexities in the use of Fz3-RFP as a reporter of Wingless activity. These observations will be of value to the Wnt field and the fly community. The revised manuscript reflects a great deal of effort to address the concerns of the first set of reviewers, and I believe that the concerns have been addressed as well as is biologically possible.

Reviewer #2: I would like to start by commending the other reviewers for their initial comments, and the authors for responding earnestly and with a manuscript that is generally very careful in presenting data, logic and conclusions.

Although I did not see the first version of the manuscript, I find that reviewer 1 has mentioned numerous key issues regarding the ovariole and FSC biology that I fully endorse as crucial. Both reviewers brought up critical genetic issues, and reviewer 2 broke the ice with regard to thinking beyond Nrt-Wg simply behaving as unable to signal at a distance. I don’t think any of these issues are fully resolved despite great efforts from the authors, and that a couple of key issues mean that the conclusions offered, at least as phrased, are not really supported.

The biggest elephant in the room is that animals with only Nrt-Wg as a genetic source of Wg in the ovary have a Fz3-FP pattern that suggests more Wnt signaling in and around FSCs rather than less. That situation would be entirely consistent with the observed reduction in cell division because others have observed that strongly increased Wnt signaling greatly reduces EdU incorporation in marked FSC clones (Reilein et al., 2017; Melamed & Kalderon, 2020). It also becomes plausible that early FCs, which normally experience almost no measurable Wnt signaling, could have a proliferation deficit (because Wnt pathway activity is increased, not decreased).

The authors likely do not favor this interpretation because it appears to conflict with their conclusions from a 2014 study. That study mainly relied on indirect measures of cell division rates, did not make strict measurements per FSC, and examined situations (altering Dlp and Mmps) where the Fz3-RFP reporter was not quantified specifically in FSCs but appears to be altered quantitatively far less than the increases reported in the current work. The inhibition of EdU incorporation that was observed by others in axn or apc mutant cells corresponds to a large increase in Wnt pathway activity in FSCs. It seems quite possible that the smaller increase seen for Mmp mutants has different proliferative consequences to the large increases due to axn, apc and Nrt-Wg in this study. It has already been shown that Wnt signaling also has a positive influence on EdU incorporation under specific conditions (when JAK-STAT signaling is reduced) (Melamed & Kalderon, 2020).

Since the Fz3-RFP result is an important phenotypic intermediate, and the authors described significant difficulty in generating rare appropriate recombinants, it would be useful (though perhaps not necessary) to examine proliferation in the very same animals (or genotypes) used for Fz3-RFP measurement.

How does the observed increased Wnt signaling over the anterior 2/3 of the germarium come about?

The authors say that direct Wg staining shows it is more restricted. The supporting data (Fig. 2G, H) do not appear to me to be very sensitive tests (would also be better for the allele over a null because that was the most informative phenotype for other properties) and it is always possible that details of fixation and staining may accentuate some Wg populations over others (not to mention that the actual molecules involved in signaling may be too few to detect). It is possible that Wg, and even Nrt-Wg, can travel in a variety of ways (exovesicles, cytonemes, transcytosis, lipoprotein particles etc.) and be released from various lipid associations at a variety of points by proteases. I personally would not feel assured from the data presented that Nrt-Wg itself is not spreading.

The authors suggest that Fz3-RFP may be increased indirectly. That is also plausible but potentially extremely complex (but is not obviously a result of loss of normal Wg spreading because it is seen for Nrt-Wg/CyO too).

It may well not be practical but the only resolution I can imagine to the question of whether Nrt-Wg itself has a shorter or longer range is to eliminate the other Wnts simultaneously.

At present, I think the statement that Nrt-Wg is tethered and acts only at short-range is only an assumption, with the Fz3-RFP results seriously challenging the assumption. My opinion is therefore that the central conclusion that includes that assumption cannot be supported. I do, however, believe that if the study were titled as an exploration of Wg range, and conclusions, were appropriately tempered on this key point, that there is a lot of useful exploration here (and that apparent contradictions between Wnt signaling and proliferation are resolved by taking the Fz3-RFP quantitative data at face value).

In my view, the Nrt-Wg studies do not resolve the issue of which Wnts are “more important” for directly signaling to FSCs in the germarium (and they add useful information for those principally interested in the subject of long-range Wg signaling in various tissues without reaching a definitive conclusion). I think that if the authors highlighted the quantitative results they found for Fz3-RFP (especially if they focus on the FSCs immediately anterior to the Fas3 border) and reducing various sources of Wnts, they could make the case that both Wg from cap cells and Wnts (mainly Wnt4) from Escort cells (collectively) contribute to the normal Wnt signaling pattern. Since it is already clear that quantitative levels of Wnt signaling affect a variety of FSC behaviors, it is to be expected that the quantitative changes observed are likely functionally significant. I believe that presentation would largely resolve prior publications on the subject and be cited as such in the future.

Some other issues:

1. Please write out actual genotypes relevant to all data panels. Despite Fig. 1 and clarification of some specific questions previously raised by reviewers, there were two or three places where I was unsure of the genotype.

2. I was unsure whether flp-out events in certain cases were just assumed or actually tested.

3. Like reviewer 1, I remain surprised that there can be normal ovariole morphology with no FSC division. Does the sequence of egg chamber maturation indicate arrest or extremely slow progress?

4. The data for Fz3-RFP in Fig. S1 seem to suggest Wnt4 is a particularly strong contributor. If correct, the text should reflect that and other appropriate quantitative impressions (most readers will not look at these data).

It is stated that for Nrt-Wg that Fz3-RFP is higher in escort cells. That is mis-leading. It is clearly higher also in FSCs and beyond. Was Fig. S2 made with Fas3 staining- ideally that should be included but I am fairly sure from the images that Fz3-RFP extends beyond escort cells.

5. The reference to previous measures of FSC mitotic clone induction rate and attempts to repeat that are, in my opinion, problematic. Those induction rates are very variable amongst animals and batches using uniform conditions and what is observed after several days depends more on the competitive nature of the FSC than the frequency of initial marking. (It is also not necessarily true that the X15-29 system requires mitosis [and not obvious that it would], according to Kirilly & Xie). I would not advise including the unpublished results distributed to reviewers.

6. I agree with reviewer 1, that PH3 staining frequency is really only informative if on a per cell basis. However, I understand the associated difficulties and for the major point being made, I think it is clear that Nrt-Wg animals have reduced PH3 labeling, though the quantitative nature of the deficit remains unknown. I remain surprised that the authors do not use EdU incorporation as a simple, albeit expensive, secondary measure (no one of these methods actually necessarily report division rates but they an both be compared to other previously measured genetic conditions).

The identity of the cells measured is important.

I would agree with the authors that useful and reliable data can be gained from measuring all cells immediately anterior to strong Fas3 staining. Even if the authors, reviewers or readers were still to believe the out-dated dogma of two FSCs, there is no way for those cells to be distinguished from the other 6 or so cells residing in exactly equivalent locations (no known marker differences; of course, entirely compatible with them all being equivalent), so counting all layer 1 FSCs (Reilein et al., 2017) will capture all “original-dogma” FSCs. My own preference would be for all FSCs to be included but I know that is technically harder to do. I also believe that doubt about conclusions from Reilein et al., 2017 should only be promulgated if the authors have an objective, named basis for doubt after careful reading of the relevant papers, including the BioRxiv deposition from the Kalderon lab.

The authors are incorrect in quoting Fadiga & Nystul with regard to FSC location, and in saying in response to reviewers that all studies place FSCs anterior to the strong Fas3 border. Fadiga & Nystul place them one cell further posterior (making the weak vs strong Fas3 border very clear, and the same as designated by others). Moreover, in Rust et al., the authors assert that all FSCs express Fas3 based on the Fadiga “result”. The alleged positioning of FSCs in those studies is indeed contrary to all prior studies, including several from the senior author and previous studies from the authors of the current study. The simplest solution is to eliminate the Fadiga citation because it does not support the (correct) assertion.

**Have all data underlying the figures and results presented in the manuscript been provided?**

Reviewer #1: Yes

Reviewer #2: Yes

PLOS authors have the option to publish the peer review history of their article (what does this mean?). If published, this will include your full peer review and any attached files.

Reviewer #1: No

Reviewer #2: No

---

## [Editor Report · Decision Letter 1]

5 Mar 2021

Dear Dr Page-McCaw,

We are pleased to inform you that your manuscript entitled "Extracellular spreading of Wingless is required for Drosophila oogenesis" has been editorially accepted for publication in PLOS Genetics. Congratulations!

Yours sincerely,

Ken M. Cadigan

Guest Editor

PLOS Genetics

Gregory P. Copenhaver

Editor-in-Chief

PLOS Genetics

Comments from the reviewers (if applicable):

**Data Deposition**

http://datadryad.org/submit?journalID=pgenetics&manu=PGENETICS-D-20-01931R1

**Press Queries**

---

## [Editor Report · Acceptance letter]

30 Mar 2021

PGENETICS-D-20-01931R1 

Extracellular spreading of Wingless is required for *Drosophila* oogenesis 

Dear Dr Page-McCaw, 

We are pleased to inform you that your manuscript entitled "Extracellular spreading of Wingless is required for *Drosophila* oogenesis" has been formally accepted for publication in PLOS Genetics! Your manuscript is now with our production department and you will be notified of the publication date in due course.

With kind regards,

Katalin Szabo

PLOS Genetics

On behalf of:
